# Influence of Condensed and Hydrolysable Tannins on the Bacterial Community, Protein Degradation, and Fermentation Quality of Alfalfa Silage

**DOI:** 10.3390/ani12070831

**Published:** 2022-03-25

**Authors:** Wencan Ke, Huan Zhang, Shengnan Li, Yanlin Xue, Yan Wang, Wencheng Dong, Yimin Cai, Guijie Zhang

**Affiliations:** 1Department of Animal Science, Ningxia University, Yinchuan 750021, China; kewc12@lzu.edu.cn (W.K.); zhang15296931276@126.com (H.Z.); 12021131211@stu.nxu.edu.cn (S.L.); wy850000750@163.com (Y.W.); dwc6716985@126.com (W.D.); 2Inner Mongolia Engineering Research Center of Development and Utilisation of Microbial Resources in Silage, Inner Mongolia Academy of Agriculture and Animal Husbandry Science, Hohhot 010031, China; xueyanlin_1979@163.com; 3Japan International Research Center for Agricultural Science (JIRCAS), Tsukuba 305-8686, Japan; cai@affrc.co.jp

**Keywords:** alfalfa, bacterial community, proteolysis, hydrolysable tannin, condensed tannin

## Abstract

**Simple Summary:**

Proteolysis widely exists in alfalfa ensiling, and up to 88% of total nitrogen could be converted to non-protein nitrogen, leading to low protein utilization efficiency. In order to limit proteolysis, tannins have been used to protect protein from microbial degradation. Although there have been studies that evaluated the efficiency of hydrolysable tannin and condensed tannin to prevent proteolysis, the two were mostly considered individually. In addition, the effects of different types of tannins on plant proteases are unknown. We turned our attention to evaluating the effects of hydrolysable tannin and condensed tannin on the bacterial community and proteolysis of alfalfa silage. In this study, both hydrolysable tannin and condensed tannin were able to reduce proteolysis in alfalfa silages—especially for condensed tannin. The application of hydrolysable tannin decreased the abundance of *Lactobacillus* and increased the abundances of *Enterococcus*, while the opposite results were observed when condensed tannin was applied. In addition, condensed tannin reduced the activity of carboxypeptidases and aminopeptidases in treated silage, whereas silage treated with hydrolysable tannin showed lower acid protease and carboxypeptidases activities. Thus, both hydrolysable tannin and condensed tannin could limit proteolysis in alfalfa silage, but the mechanism and their effects on the bacterial communities were different.

**Abstract:**

This study evaluated the effects of hydrolysable tannin (HT) and condensed tannin (CT) on the bacterial community, fermentation quality, and proteolysis of alfalfa silage. Alfalfa was wilted to a dry matter (DM) of 35% fresh weight and ensiled with or without 4% HT or 4% CT. The application rates of tannins were based on fresh weight, and each treatment was ensiled in triplicate. After 60 d of fermentation, the CT-treated group had lower concentrations of ammonia nitrogen (NH_3_-N) and free amino acid nitrogen (AA-N), but greater lactic acid concentration, than those in the control and HT-treated silage (*p* < 0.05). Compared to the control group, the application of tannins increased the abundance of *Pseudomonas* (negatively correlated with aminopeptidases activity), and decreased the abundance of *Pediococcus*—which was positively correlated with aminopeptidases activity—and the concentrations of non-protein nitrogen (NPN), NH_3_-N, and AA-N. The application of HT decreased the abundance of *Lactobacillus* and increased the abundances of *Enterococcus*, while the opposite results were observed in the CT-treated group. The application of HT and CT reduced the proteolysis in treated silages, but the two were different in terms of their mechanism and their effects on the bacterial communities of the alfalfa silage.

## 1. Introduction

Proteolysis widely exists in alfalfa ensiling, and approximately 50–80% of the total nitrogen in alfalfa silages could be converted to non-protein nitrogen (NPN) [1], leading to low protein utilization efficiency [2]. In order to limit proteolysis in silage fermentation, inoculants and organic acids are used to stimulate fermentation of ensiled forage, and several studies have confirmed their positive effects in decreasing proteolysis [3,4].

Tannins are a diverse group of polyphenols with the capacity to bind proteins, protecting them from microbial degradation [5]. These protein-binding properties suggest the potential of tannins as silage additives to reduce proteolysis during silage-making [6,7]. Unlike organic acids and inoculants used during ensiling, the broad-spectrum antimicrobial properties of tannins could decrease the activity of bacterial enzymes by precipitating them and binding their cell membranes, thereby reducing the growth of undesirable microorganisms [8]. Commercial tannins could be classified into two main groups: hydrolysable tannin (HT) and condensed tannin (CT). Compared with CT, HT is not commonly used in silages, due to its negative effects on animal performance [9]. However, recent studies have shown that HT could also increase nitrogen utilization without diverse effects on animal performance [10]. Li et al. [11] have also confirmed that HT is effective in limiting protein degradation in alfalfa silage. Although there have been studies evaluating the efficiency of HT and CT to prevent proteolysis [12,13], in most of them their evaluation was individual, and the comparison of the effects of these two tannins on proteolysis is still unknown. In addition, most proteolysis results from plant proteases [14], such as carboxypeptidases, aminopeptidases, and acid proteinases [15]; however, very few studies have compared the efficiency of these two kinds of tannins on proteolysis, or their effects on the activities of plant proteases.

Silage fermentation is a dynamic process involving the succession of the bacterial communities and changes in metabolites. Revealing the bacterial community of silage is vital for ensuring silage preservation. Thus far, most studies have shown that studying the silage microbiome and defining microbial communities with a degree of detail [16,17] could improve our understanding of the biological processes involved in silage. Although Colak et al. [18] have proven that tannins can reduce microbial activity during ensiling, to the best of our knowledge, the effects of CT and HT on the bacterial community have rarely been studied.

Thus, it is hypothesized that both CT and HT could limit proteolysis during silage fermentation, but their effects and mechanism are different. Therefore, this study investigated the effects of condensed and hydrolysable tannins on the bacterial community structure, protein degradation, and fermentation quality of alfalfa silage.

## 2. Materials and Methods

### 2.1. Fresh Alfalfa and Ensiling

Alfalfa (cultivar “Gannong No. 4”) was cultivated in the field at Ningxia University (longitude 106°03′ E, latitude 38°33′ N, altitude 1135 m; Yinchuan City, China) and harvested at the early blooming stage from three randomly selected plots (12 m × 9 m). Fresh alfalfa was naturally wilted to a dry matter (DM) of approximately 35% (~15 h) and chopped into 1–2 cm fragments.

Chopped forages from three plots were thoroughly mixed and randomly divided into 12 piles, of which 3 piles were stored at −20 °C as fresh samples and the remaining 9 piles were ensiled according to the following treatments: (1) control (no tannin); (2) HT (65% hydrolysable tannin, OENOTANNIN MIXTE MG, Yantai Diboshi Self-Brewing Co., Ltd., Yantai, China); or (3) CT (65% condensed tannin; VINITAN, Lamothe-Abiet, Bordeaux, France). A total of 61.5 g of tannin powder was dissolved in 100 mL of distilled water to prepare a tannin solution (0.4 g/mL). A total of 10 mL of HT or CT solution was mixed with 1 kg of fresh alfalfa through separate spray bottles, and the same amount of distilled water was applied to the control group. Approximately 500 g of mixed material was ensiled in a polyethylene plastic bag silo (300 × 270 mm; Embossed Food Saver Bag Co., Ltd., Chengdu, China) and vacuumed with a sealer (DZ-400, Shandong Zhucheng Yizhong Machinery Co., Ltd., Zhucheng, China). The silages were stored at room temperature (20–26 °C) for 60 days.

### 2.2. Fermentation Characteristics and Chemical Composition

After 60 d of ensiling, the bags were opened, and a portion of silage was immediately frozen (−20 °C) for further analysis. A 20 g sample of each silage was mixed with distilled water (1:9) and homogenized for 2 min, and then filtered through four layers of cheesecloth. After that, the pH of the filtrate was immediately measured with a pH meter (S-3G, Shanghai INESA Scientific Instrument Co., Ltd., Shanghai, China). The concentrations of lactic acid (LA), acetic acid (CA), propionic acid (PA), and butyric acid (BA) were analyzed via high-performance liquid chromatography (HPLC, KC-811 column, Shodex; Shimadzu, Kyoto, Japan; oven temperature 50 °C; flow rate 1 mL/min; SPD 210 nm). A portion of the filtrate was mixed with 250 g/L (*w*/*v*) trichloroacetic acid at a ratio of 4:1 and stored at 4 °C overnight to precipitate the protein. After centrifugation at 18,000× *g* for 15 min at 4 °C, the supernatant was collected to determine the non-protein nitrogen (NPN), free amino acid nitrogen (AA-N), and ammonia nitrogen (NH_3_-N) contents [19,20]. The acid protease, carboxypeptidase, and aminopeptidase activities of the alfalfa silage were determined according to the methods of Guo et al. [21]. Enzymatic activity was expressed as units per hour on a DM basis (unit h^−1^ g^−1^ DM).

The silage samples (10 g) were mixed with distilled water (90 mL), shaken for 30 min in a constant-temperature incubator shaker (ZHLY-180F, Shanghai Zhichu Instrument Co., Ltd., Shanghai, China). Serial dilutions (10-fold) were carried out with sterile saline solution, and microorganisms were cultured in different media. Lactic acid bacteria (LAB), yeasts (molds), and coliform bacteria were cultured using Rogosa agar (HB0384), potato dextrose agar (HB0233), and eosin–methylene blue agar (CM105), respectively. The LAB were incubated under anaerobic conditions at 30 °C for 48–72 h, while the yeasts (molds) and coliform bacteria were incubated under aerobic conditions at 30 °C for 48–120 h. The microbial counts were converted to log_10_ colony-forming units (cfu) per gram on a fresh matter (FM) basis.

The DM contents of fresh forages and silages were analyzed by drying samples in a forced-air oven at 65 °C for 72 h. Dried samples were ground with a mill (a 1 mm screen) and used to determine the contents of crude protein (CP), water-soluble carbohydrate (WSC), neutral detergent fiber (NDF), and acid detergent fiber (ADF). The CP was calculated by multiplying total N by 6.25 [22]. The WSC concentration was determined according to the method of Thomas [23]. The NDF and ADF contents were quantified using the procedures of Van Soest et al. [24] and Goering and Van Soest [25], using an ANKOM A2000i fiber analyzer (A2000i, ANKOM Technology, New York, NY, USA).

### 2.3. Bacterial Community Composition

The DNA extraction of wilted alfalfa and silages was performed with a DNA isolation kit (DP712, Tiangen Biochemical Technology Ltd., Beijing, China) according to the manufacturer’s instructions. The quality of extracted DNA samples was tested using a spectrophotometer (Thermo Fisher Scientific, Waltham, MA, USA). The PCR amplification of the bacterial 16S rRNA genes was performed using the forward primer 338F (5′-ACTCCTACGGGAGGCAGCAG-3′) and the reverse primer 806R (5′-GGACTACHVGGGTWTCTAAT-3′). The reaction system and PCR parameters were the same as described by Yang et al. [26].

The PCR samples were purified with Agencourt AMPure Beads (Beckman Coulter, Indianapolis, IN, USA) and then sequenced using the Illumina MiSeq platform (Majorbio, Shanghai, China).

### 2.4. Statistical Analysis

The fermentation parameters, chemical composition, nitrogen components, and protease activity of alfalfa silage were analyzed with the Statistical Package for the Social Sciences (SPSS Version 19.0, SPSS Inc., Chicago, IL, USA) using one-way analysis of variance (ANOVA), and multiple comparisons were performed using Tukey’s HSD test. Significance was declared at *p* < 0.05.

Clustering was carried out in operational taxonomic units (OTUs) with 97% similarity using USEARCH software (UPARSE algorithm). Chloroplast and mitochondrial reads, along with reads shorter than 50 bp, were removed, and the OTU counts were rarefied to an equal number of sequences per sample before calculating diversity indices using MOTHUR (version 1.30.2). The minimum number of sequences across all of the samples was 42,047. The sequences were compared against the Silva (SSU123) 16S rRNA database, and a confidence threshold of 70% was used to obtain the composition of each sample. The data were analyzed using the online platform of the Majorbio Cloud Platform (www.majorbio.com, accessed on 16 January 2022).

## 3. Results

### 3.1. The Chemical Composition and Epiphytic Microflora of Wilted Alfalfa

The chemical composition and epiphytic microflora of wilted alfalfa are presented in Table 1. The counts of LAB and yeast were 4.01 and 3.86 log_10_ cfu/g FM, respectively. Coliform bacteria and molds were undetectable. Alfalfa was wilted to a DM of 33.3% FM and the CP, WSC, NDF, and ADF contents in wilted alfalfa were 22.8, 1.79, 41.9, and 30.0, respectively.

### 3.2. Silage Bacterial Community

The indices of alpha diversity in the alfalfa silages are shown in Table 2. The number of circular consensus sequences ranged from 47,608 to 51,563. Compared with the control silage, HT- and CT-treated silages had numerically higher Shannon indices and numerically lower Chao1 indices, but not to a statistically significant extent. The coverage indices in all silages were higher than 0.99.

Differences in the bacterial communities of alfalfa silage identified by principal coordinate analysis (PCoA, Figure 1a) based on Bray–Curtis distances are illustrated in Figure 1. According to PCoA analysis, the bacterial communities showed distinct separation between the three treatments. PC1 and PC2 accounted for 74.76% and 18.38% of the total variance, respectively. There were 98, 128, and 120 OTUs in the control, HT-, and CT-treated silages, respectively, among which 21, 40, and 33 were unique, respectively (Figure 1b). In addition, 50 OTUs were shared by all silages.

The bacterial communities of the alfalfa silages were mainly composed of two phyla and seven genera (Figure 2). The control silage had a higher relative abundance of *Firmicutes* and lower abundance of *Proteobacteria* than tannin-treated silages (Figure 2a). At the genus level (Figure 2b), the dominant bacteria in the control group were *Enterococcus* (55.17%), *Lactobacillus* (27.50%), and *Pediococcus* (11.7%). The application of HT increased the abundance of *Enterococcus* (73.68%) and decreased the abundance of *Lactobacillus* (18.48%). The highest proportions of *Lactobacillus* (50.04%) and *Enterococcus* (41.25%) were observed in the CT-treated silage.

### 3.3. Fermentation Characteristics and Chemical Composition of Alfalfa Silages Ensiled for 60 d

The fermentation profiles and chemical composition of alfalfa silages are shown in Table 3. The application of HT and CT reduced lactic acid bacteria, especially when HT was applied. Both HT and CT also inhibited the growth of yeasts, but more drastically in CT-treated silage. The application of CT resulted in a lower pH value and greater concentration of lactic acid when compared to the control group, whereas the opposite result was observed when HT was applied. Both HT and CT reduced the concentration of acetic acid in treated groups compared to the control group.

The application of CT and HT had no effects on the contents of DM, NDF, ADF, or CP, but limited proteolysis during silage fermentation. Compared with the control group, HT- and CT-treated silages presented lower concentrations of NPN, AA-N, and NH_3_-N, and the result was more drastic for the concentrations of AA-N and NH_3_-N in the CT-treated group (*p* < 0.05).

Compared with the control group, CT reduced the activity of carboxypeptidases and aminopeptidases in treated silage, whereas HT-treated silage had lower activity of acid proteases and carboxypeptidases.

### 3.4. Correlation between the Microorganism and Fermentation Parameters

The relationships between the microorganisms (at the genus level) and the fermentation parameters in alfalfa silage were analyzed via Spearman’s correlation analysis (Figure 3). There was a positive relationship between *Weissella* and acetic acid (*p* < 0.01) and NPN (*p* < 0.05). *Pediococcus* was positively associated with NPN (*p* < 0.05), NH_3_-N (*p* < 0.01), and AA-N (*p* < 0.01). Aminopeptidase activity was negatively correlated with *Lactobacillus* (*p* < 0.05) and *Pseudomonas* (*p* < 0.01), but positively correlated with *Pediococcus* (*p* < 0.05).

## 4. Discussion

### 4.1. Diversity and Composition of Bacterial Community

In this study, all treatments had a coverage index higher than 0.99, which means that the sequencing data were sufficient to reliably analyze the bacterial community. The Shannon and Chao1 indices are used to reflect bacterial community diversity and richness, respectively [27]. In this study, HT- and CT-treated silages had numerically lower bacterial diversity and numerically higher bacterial richness when compared to the control group, which may be due to the bacteriostatic activity of tannin [12]. The distinct separation between the control and tannin-treated silages in the PCoA results suggests that tannins alter the bacterial community of alfalfa silage. The Venn diagram showed that HT and CT promoted the growth of some microorganisms of alfalfa silage, as indicated by the greater numbers of total and unique OTUs, which may be attributed to the biological activity caused by the exogenous addition of tannins [28,29]. 

Hu et al. [30] observed that *Proteobacteria* were the preponderant bacteria in alfalfa before ensiling, and there was a tendency towards a shift from *Proteobacteria* to *Firmicutes* during ensiling. In this study, the addition of tannins increased the relative abundance of *Proteobacteria* and decreased the relative abundance of *Firmicutes* in alfalfa silage. This result might be explained by the proliferation of LAB being affected by the antibacterial activity of tannins [31]. In this study, *Enterococcus* dominated the fermentation process in the control and HT-treated silages—especially in HT-treated silages. This result is consistent with previous investigations by Ni et al. [32], in which *Enterococcus* accounted for a large proportion of bacteria in wilted alfalfa silage. Compared with the control silage, lower *Lactobacillus* and higher *Enterococcus* abundances were observed in HT-treated silage, while *Lactobacillus* increased by 22.49% and *Enterococcus* decreased by 13.83% in the CT-treated silage. It seems that LAB of different genera respond to tannins differently. Tabasco et al. [33] also showed that there were differences in the growth of LAB species in the presence of condensed tannin.

### 4.2. Fermentation Characteristics of Alfalfa Silage

Typically, tannins have negative effects on the growth of LAB [34,35]. Compared with the control silage, the LAB populations decreased from 6.74 to 5.57 and 6.57 log_10_ cfu/g FM in HT- and CT-treated silages, respectively. A previous study showed that HT has a greater ability to inhibit the growth of LAB than CT during ensiling [13], which is consistent with the lower LAB population in HT-treated silage.

Silage pH is an indicator to evaluate the quality of silage. In this study, CT-treated silage had the lowest pH and greater LA when compared to the other silages. This was probably due to the greater abundance of *Lactobacillus* in CT-treated silage. *Lactobacillus* species have a greater tolerance to acidic environments than *Enterococcus*, *Pediococcus*, or *Weissella* [26], indicating that a greater *Lactobacillus* abundance in silage is likely beneficial for decreasing pH during ensiling. Cavallarin et al. [36] found that the pH of alfalfa silage treated with HT was 5.4, which was similar to the results of this study. The high pH and the lower LA concentration in the HT-treated silage may be connected with the lower abundance of *Lactobacillus*.

The relative abundances of bacteria are highly correlated with silage quality [27]. *Weissella* is a taxon of heterofermentative LAB that produces acetic acid, LA, and CO_2_ [37], which is consistent with the significantly positive relationship between the relative abundance of *Weissella* and acetic acid production in this study.

### 4.3. Nitrogen Components and Protease Activity

In legume silages, *Clostridia* typically cause strong proteolysis and increase the levels of soluble nitrogen and NH_3_-N [38]. Tannins have the ability to bind proteins, and have been reported to reduce NPN and NH_3_-N concentrations during silage fermentation [36]. In this study, HT and CT reduced the concentrations of NPN, NH_3_-N, and AA-N in treated silages In addition, CT has a stronger protein-binding capacity than HT, because the higher molecular weight of condensed tannins promotes cross-linking with proteins [39], which may led to lower concentrations of NH_3_-N and AA-N in CT-treated silages.

Association analysis is a simple and practical analysis technique, which consists of discovering the association or correlation existing between a large number of datasets, thus describing the laws and patterns of the simultaneous occurrence of certain attributes in a thing. Spearman’s correlation analysis indicated that *Weissella* and *Pediococcus* interacted with NPN, NH_3_-N, and AA-N. Liu et al. [40] found that *Weissella* and *Pediococcus* rapidly proliferated at the early stage of fermentation, and then decreased under subsequent acidic condition. In this study, *Pediococcus* and *Weissella* were positively associated with NPN, while *Pediococcus* was positively associated with NH_3_-N and AA-N. These results show that tannins may change the adaptable environment for microorganisms in silage, leading to lower relative *Weissella* and *Pediococcus* abundances and lower proteolysis.

In this study, the activity of protease was inhibited in the alfalfa silages treated with tannins, facilitating protein preservation. Compared with the control silage, the CT-treated silage exhibited decreased activity of carboxypeptidases, which may be due to the lower pH in CT-treated silage, as carboxypeptidase activity rapidly decreases at pH 4.5 [41]. However, the decrease in carboxypeptidase activity in the HT-treated silage was probably attributable to the inhibition of protease action by tannin-binding proteins. Spearman’s correlation analysis between protease activity and the bacterial community confirmed a positive association between the activity of aminopeptidases and *Pediococcus* abundance, but a negative correlation with *Lactobacillus* and *Pseudomonas* abundance. This may be because *Lactobacillus* can promote pH decline and impede protease activity. Ogunade et al. [42] illustrated that a negative correlation was found between relative *Pseudomonas* abundance and NH_3_-N concentration. *Pseudomonas* may retain the protein concentration and reduce the NH_3_-N concentration and yeast population of silage [27]. In this study, *Pseudomonas* had a significant effect on the activity of aminopeptidases. The higher abundance of *Pseudomonas* in the HT- and CT-treated silages might contribute to protein preservation by inhibiting aminopeptidase activity.

## 5. Conclusions

Both HT and CT could reduce proteolysis in alfalfa silages, but more drastically in CT-treated silage. The application of HT decreased the abundance of *Lactobacillus* and increased the abundance of *Enterococcus*, while the opposite results were observed when CT was applied. In addition, CT reduced the activity of carboxypeptidases and aminopeptidases in treated silage, whereas silage treated with HT had lower activity of acid proteases and carboxypeptidases. Thus, both HT and CT could limit proteolysis in alfalfa silage, but the mechanism and their effects on the bacterial communities were different.

## Figures and Tables

**Figure 1 animals-12-00831-f001:**
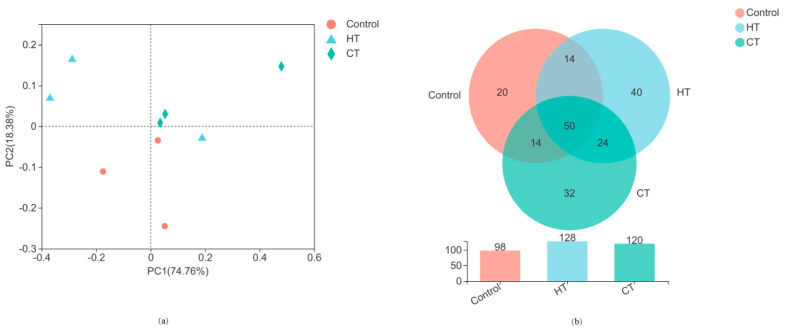
Analysis of the bacterial community of alfalfa silage at the operational taxonomic unit (OTU) level, based on principal coordinate analysis (PCoA, (**a**)) and a Venn diagram (**b**). HT = hydrolysable tannin; CT = condensed tannin.

**Figure 2 animals-12-00831-f002:**
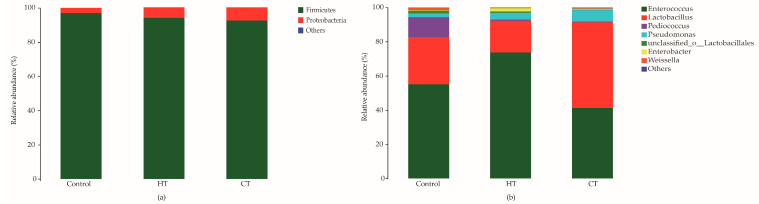
The structure of the bacterial community in silages at the phylum (**a**) and genus levels (**b**). HT = hydrolysable tannin; CT = condensed tannin.

**Figure 3 animals-12-00831-f003:**
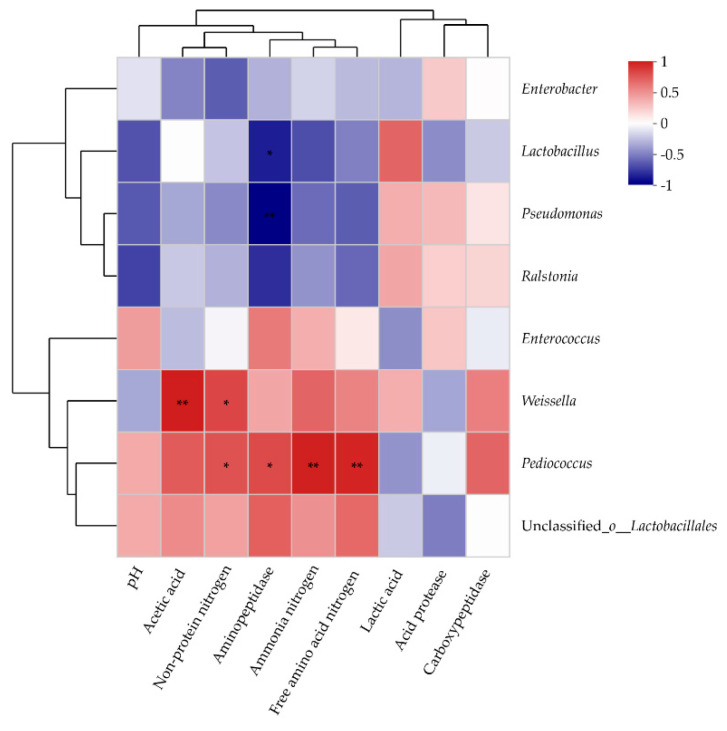
The correlation between microorganisms and fermentation parameters at the genus level using Spearman’s correlation analysis. Correlation R values and *p*-values were obtained through calculation. R values are shown in different colors in the figure. If *p*-values are less than 0.05, they are marked with as follows: * 0.01 < *p* ≤ 0.05, ** 0.001 < *p* ≤ 0.01.

**Table 1 animals-12-00831-t001:** Characteristics of alfalfa prior to ensiling (means ± standard deviation).

Item ^1^	Alfalfa
LAB (log_10_ cfu/g of FM)	3.86 ± 0.11
Yeasts (log_10_ cfu/g of FM)	4.01 ± 0.05
Coliform bacteria (log_10_ cfu/g of FM)	-
Molds (log_10_ cfu/g of FM)	-
DM (%)	33.3 ± 1.06
CP (% DM)	22.8 ± 0.21
WSC (% DM)	1.79 ± 0.12
NDF (% DM)	41.9 ± 2.48
ADF (% DM)	27.0 ± 0.68

^1^ LAB = lactic acid bacteria; FM = fresh matter; DM = dry matter; CP = crude protein; WSC = water-soluble carbohydrate; NDF = neutral detergent fiber; ADF = acid detergent fiber; cfu = colony-forming unit.

**Table 2 animals-12-00831-t002:** Alpha diversity of alfalfa silage.

Item ^1^	Alfalfa Silage ^2^			SEM ^3^	*p*-Value
	Control	HT	CT		
CCS	52,740	52,467	48,431	3382	0.62
Shannon index	1.24	0.69	0.99	0.15	0.11
Chao1 index	83.3	120	110	24.7	0.58
Coverage index	0.99	0.99	0.99	0.00	0.62

^1^ CCS = circular consensus sequence. ^2^ HT = hydrolysable tannin; CT = condensed tannin. ^3^ SEM = standard error of the mean.

**Table 3 animals-12-00831-t003:** Microbial population, fermentation parameters, chemical composition, nitrogen components, and protease activity of alfalfa silage.

Item ^1^	Alfalfa Silages ^2,3^			SEM ^4^	*p*-Value
	Control	HT	CT		
Microbial population					
LAB (log_10_ cfu g^−1^ of FM)	6.74 ^a^	5.57 ^c^	6.41 ^b^	0.05	<0.01
Yeast (log_10_ cfu g^−1^ of FM)	4.08 ^a^	3.80 ^b^	3.53 ^c^	0.05	<0.01
Coliform bacteria (log_10_ cfu g^−1^ of FM)	-	-	-	-	-
Molds (log_10_ cfu g^−1^ of FM)	-	-	-	-	-
Fermentation parameters					
pH	4.86 ^b^	5.06 ^a^	4.61 ^c^	0.02	<0.01
Lactic acid (% DM)	2.83 ^b^	2.14 ^c^	3.21 ^a^	0.04	<0.01
Acetic acid (% DM)	2.28 ^a^	1.99 ^b^	2.06 ^b^	0.02	<0.01
Propionic acid (% DM)	0.67	-	-	-	-
Butyric acid (% DM)	-	-	-	-	-
Chemical composition					
DM (%)	32.7	32.3	33.3	0.56	0.48
NDF (% DM)	40.3	39.9	41.3	0.58	0.30
ADF (%DM)	27.3	29.9	28.3	2.87	0.97
Nitrogen components					
CP (% DM)	21.7	21.5	21.5	0.26	0.78
NPN (% TN)	73.1 ^a^	51.4 ^b^	50.4 ^b^	24.14	<0.01
NH_3_-N (% TN)	11.5 ^a^	8.32 ^b^	5.71 ^c^	0.17	<0.01
AA-N (% TN)	31.1 ^a^	20.3 ^b^	16.8 ^c^	4.25	<0.01
Protease activity					
Acid proteases (unit h^−1^ g^−1^ DM)	7.99 ^a^	6.39 ^b^	8.07 ^a^	0.18	<0.01
Carboxypeptidases (unit h^−1^ g^−1^ DM)	17.6 ^a^	8.53 ^b^	8.35 ^b^	1.76	0.02
Aminopeptidases (unit h^−1^ g^−1^ DM)	24.0 ^a^	24.4 ^a^	15.8 ^b^	0.67	<0.01

^1^ LAB = lactic acid bacteria; DM = dry matter; NDF = neutral detergent fiber; ADF = acid detergent fiber; CP = crude protein; NPN = non-protein nitrogen; TN = total nitrogen; NH_3_-N = ammonia nitrogen; AA-N = amino acid nitrogen; cfu = colony-forming unit; FM = fresh matter. ^2^ HT = hydrolysable tannin; CT = condensed tannin. ^3 a–c^ indicate a significant difference within a row (*p* < 0.05). ^4^ SEM, standard error of the mean.

## Data Availability

All of the sequence data of this study were deposited in the NCBI Sequence Read Archive (SRA) database under the accession number SRP287063.

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
