# Peer review of "Influence of Condensed and Hydrolysable Tannins on the Bacterial Community, Protein Degradation, and Fermentation Quality of Alfalfa Silage"

_animals, 2022, doi:10.3390/ani12070831_

Round 1

Reviewer 1 Report

  1. Line -12 Remove the word "simple" before the word "Summary".
  2. Line -15 Add "that" before the word evaluated.
  3. Line-17 Delete the word "activities"
  4. Line-17 Attentions should be replaced by "attention" 
  5. Line-51 Replace "protein" with "proteins''
  6. Line-55 Replace "reduced" with "reducing"
  7. Line-56 in to should be written as a single word "into"
  8. Line-57 Replace "evaluated" with "evaluating"
  9. Lines- 59-60 Write plurals like "carboxypeptidases, aminopeptidases, acid proteinases" instead of "carboxypeptidase, aminopeptidase, acid proteinase"
  10. Line-60 Little literature can be replaced by "very few studies"
  11. Line-68-69. "To our best knowledge, the effects of condensed and hydrolysed tannins 68 on the bacterial community have rarely studied." Rewrite this sentence, this is poorly written. 
  12. Line-97 insert the word "that" in the sentence "After that the pH of filtrate was"
  13. Line-113 replace "was" with "were"
  14. Line-128 replace "were" with "was"
  15. Line-130 remove extra T from the word The. typing error

INTRODUCTION IS POORLY WRITTEN , LOT OF GRAMMATICAL ERRORS, IT SHOULD BE RE-WRITTEN. 

OVERALL A GOOD STUDY. 

Reviewer 2 Report

The authors studied the effect of the inclusion of tannins in the procces of silage in order to reduce the protein degration. They provide an interesting contribution to the quality of alfalfa silage.

Please consider the following suggestions:

-In material and methods is not clear the origin of the samples as you had 3 plots. You divided the samples, did you take one sample per plot per each treatment?

L87: by via separete, Is it written correctly?

L97: add a comma after "After,"

L121: Delete one to (it is repeated)

L130: Change TThe by The

Table 1: I recommend expressing the chemical composition in g/kg DM as is most accepted in scientific papers. In addition the   the standard desviation in all parameters presented in Table 1 should be appear.

L167-168: It seems that the Chao 1 indices also have a trend to increase, rewrite this sentence. In fact, no tendency has observed as the P was 0.11, so please rewrite these results.

L190: Delete a comma after (18.48%).

L186-187: Did you compare the relative abundance of the phyla statistically?

L199: Change table 2 by table 3

L202-204: Move the parragraph regarding the microbial population before explaning the fermentation parameters, in  to follow the same order that are presented in the table 3.

L207: Change "were lower in the concentrations of..." by " presented lower concentration of....

L238-240: It is not true, no statistical difference were found, therefore you should discuss in different way, or write that only numerical differences were found.

L267: change "to other" by "to the other"

Reviewer 3 Report

The authors conducted a study on "Influence of Condensed and Hydrolysed Tannins on the Bacterial Community, Protein Degradation, and Fermentation Quality of Alfalfa Silage"

The manuscript is interesting, however for it to be processed it needs major revision.

The introduction of MS seems very simple to me and I believe that the authors can improve and deepen the hypothesis presented.

In materials and methods, it was not possible to observe at this moment how the chemical composition mentioned in item 3.1 was performed.

The multivariate statistical analysis I do not know if would be the most appropriate, as I do not see a very large number of variables to be analyzed, the authors could better justify this in materials and methods. Why do authors use this statistical method? Also, no support references!

The discussion is dynamic, I believe it is within the MS proposal.

The conclusion needs to be improved, as I have not seen in depth the authors can improve their main findings and highlights in the manuscript, without using parts of the work that have already been previously discussed in other sections.

I hope this is contributing positively to improving the quality of the manuscript, I apologize if I was too judicious.

Round 2

Reviewer 1 Report

Line 55, (2)should be written as two

Lines 111-112, The acid proteases, carboxypeptidases and aminopeptidases activities

(it should be The acid protease, carboxypeptidase and aminopeptidase activities)

Reviewer 2 Report

The authors made all the suggestions. I accept the manuscript in the current form.

Reviewer 3 Report

All authors as recommended revisions the manuscript can be accepted for publication